# Evaluation of the added value of Brain Natriuretic Peptide to a validated mortality risk-prediction model in older people using a standardised international clinical assessment tool

John W. Pickering[1,2], Richard Scrase[3], Richard Troughton[2], Hamish A. Jamieson[1]*

1 Better Ageing with Big Data Research Group, Department of Medicine, University of Otago, Christchurch, New Zealand, 2 Department of Medicine, University of Otago, Christchurch Heart Institute, Christchurch, New Zealand, 3 Te Whatu Ora–Health New Zealand, University of Otago, Christchurch, New Zealand

* hamish.jamieson@otago.ac.nz

**Data Availability Statement:** Because of restrictions of use of clinical data we cannot make the data set freely available. However, we can make

## Abstract

The ability to accurately predict the one-year survival of older adults is challenging for clinicians as they endeavour to provide the most appropriate care. Standardised clinical needs assessments are routine in many countries and some enable application of mortality prediction models. The added value of blood biomarkers to these models is largely unknown. We undertook a proof of concept study to assess if adding biomarkers to needs assessments is of value. Assessment of the incremental value of a blood biomarker, Brain Naturetic Peptide (BNP), to a one year mortality risk prediction model, RiskOP, previously developed from data from the international interRAI-HomeCare (interRAI-HC) needs assessment. Participants were aged ≥65 years and had completed an interRAI-HC assessment between 1 January 2013 and 21 August 2021 in Canterbury, New Zealand. Inclusion criteria was a BNP test within 90 days of the date of interRAI-HC assessment. The primary outcome was one-year mortality. Incremental value was assessed by change in Area Under the Receiver Operating Characteristic Curve (AUC) and Brier Skill, and the calibration of the final model. Of 14,713 individuals with an interRAI-HC assessment 1,537 had a BNP within 90 days preceding the assessment and all data necessary for RiskOP. 553 (36.0%) died within 1-year. The mean age was 82.6 years. Adding BNP improved the overall AUC by 0.015 (95% CI:0.004 to 0.028) and improved predictability by 1.9% (0.26% to 3.4%). In those with no Congestive Heart Failure the improvements were 0.029 (0.004 to 0.057) and 4.0% (0.68% to 7.6%). Adding a biomarker to a risk model based on standardised needs assessment of older people improved prediction of 1-year mortality. BNP added value to a risk prediction model based on the interRAI-HC assessment in those patients without a diagnosis of congestive heart failure.

it available if contacted subject to ethics approval from a New Zealand ethics board and a formal use agreement between ourselves and those who wish to use the data. For New Zealand data this is particularly important because some of the data is from Māori, the indigenous people of New Zealand, who are acknowledged to have sovereignty of their data and its use. The restrictions are imposed by the New Zealand Ministry of Health and Disability Ethics Committee (HDEC). Data access queries may be sent to the Department of Medicine, University of Otago Christchurch, Christchurch, New Zealand (contact via christchurch@otago.ac. nz).

**Funding:** Health Research Council of New Zealand grant 17/363 The funders had no role in study design, data collection and analysis, decision to publish, or preparation of the manuscript.

**Competing interests:** The authors have declared that no competing interests exist.

## Introduction

At a global level the population is ageing and western countries in particular are doing so with increasing levels of comorbidities and complex health conditions [1]. To be able to provide timely and appropriate care for older individuals with complex conditions, clinicians need reliable mortality prediction tools to aid both clinical decision making and inform conversations with patients and family. In this way, wherever possible, common understanding can be reached about appropriate interventions and treatment options including making comprehensive future health care plans [2].

Reliable and accurate mortality prediction can be of significant value to clinicians when considering the risks and benefits of specific medications or when formulating individualised care plans for older adults with considerable variation in health status and disability [3–5]. The development of mortality tools which are more easily applied in the real-world environment and which can provide clinically meaningful information to guide mortality assessment have been facilitated by the evolution of standardised assessment tools and the accompanying big data [6]. The international Residential Assessment Instrument (interRAI) is one such suite of standardised assessment tools containing questions around demographics, function, comorbidities, and living conditions, and which is now used in over 35 countries and in a multitude of clinical settings [7,8]. Each tool is specific to a setting, e.g. long-term care facilities, or discipline e.g mental health, although a large proportion of the questions are identical. This enables monitoring of changes of status between assessments as the patient moves through their health journey and life course [9]. Although the primary purpose of any interRAI assessment is to provide a standardised assessment to create individualised care plans, they also provide a comprehensive dataset at a population level which can be valuable to aid research aimed at improving outcomes for our most vulnerable older population [10].

The interRAI-Home Care (interRAI-HC) is mandated for use throughout all of New Zealand with older adults requiring assessment for Government funded support or who are being considered for possible entry into aged residential care [8]. This means the individuals being assessed tend to have complex co-morbidities and at a population level are frailer and more vulnerable than their peers. This clinically vulnerable group is, therefore, the exact older cohort clinicians would most want to have a greater understanding of their morbidity trajectory, so that they can more appropriately support with care better tailored to the individual.

The authors recently developed and validated a mortality risk model (RiskOP) for those people aged 65 plus using data from the interRAI-HC only [11]. The RiskOp model predicts one-year mortality with good discrimination and excellent calibration.

The broad context for this research is the question of whether individual biomarkers, not currently part of most needs assessments, can improve risk prediction models that are based on standardised assessments in older adults. To answer this question, we undertook a proof of concept using a validated prediction model with good discrimination and calibration (therefore with less room for improvement than a poor one), RiskOp and with a biomarker associated with mortality across a variety of clinical settings, Brain Natriuretic Peptide (BNP) [12,13]. In western countries in particular heart failure is a leading cause of death among the older population [14]. BNP is a good predictor of mortality in adults both with and without a prior diagnosis of heart failure [13,15,16]. We also chose BNP because it is in frequent clinical use, it is inexpensive and it is clinically accessible. We hypothesised that BNP may add useful prognostic information to the RiskOP one-year mortality risk prediction model for older adults.

## Methods

### Design

We assessed the added value of BNP to the RiskOP one-year mortality prediction model which uses the interRAI-HC instrument [11]. The interRAI-HC is administered to older people to assess their need for home support or for entry to a long-term care facility. The inteRAI -HC assessment tool is administered by a trained health professional who is usually either a Registered Nurse or a Social Worker. These health professionals are required to attend a training course and are then need to complete yearly updates to ensure that the quality of their assessments is maintained. These assessments consist of over 200 questions across 20 domains asked of the patient and sometimes their care givers. Data quality and completeness has been previously demonstrated [8]. RiskOP uses the answer to 16 questions concerning Age, Sex, Body Mass Index, Cancer, Congestive Heart Failure (CHF), Previous Stroke, Parkinsons, Dyspnea, Weight loss, Fatigue, Oxygen therapy, Skin ulcer, frequency of going out, exercise, time since last hospital stay and change in activities of daily living.

### Participants

Participants included all people aged ≥65 years who completed an interRAI-HC assessment between 1 January 2013 and 21 August 2020 in Canterbury, New Zealand, and who had given written consent for their data to be used for research. Previous research has demonstrated 93% of all those who are assessed give this consent [8]. The RiskOP model had been developed in a subset of these patients (up to March 2018) [11]. Ethical approval was provided by the New Zealand Ministry of Health and Disability Ethics Committee (HDEC) (14/STH/140/AM08).

Inclusion criteria was a single BNP test within 90 days of the date of interRAI-HC assessment. Exclusion criteria was those identified as being on a palliative pathway. This was identified by a question which asked if they had been told that in the best clinical judgment of the physician the individual has end stage disease with approximately 6 or fewer months to live.

The interRAI-HC assessment data and BNP test results was provided by the Canterbury District Health Board. The National Mortality Collection Register administered by the New Zealand Ministry of Health provided the dates of deaths. Linkage between the data sets was made by encrypted national health index numbers as every person who has ever interacted with the public health system in New Zealand has a unique number. All participants were followed for a minimum of 12 months or until death.

### Subgroups

Participants with and without a diagnosis of Congestive Heart Failure separately.

### Outcomes

The primary outcome was one-year mortality.

### Statistical analysis

Data are presented as n (%) for categorical variables, mean and standard deviation for normally distributed quantitative variables and median (lower quartile and upper quartile) for non-normally distributed variables. All confidence intervals are 95% calculated using bootstrapping.

The RiskOP model was applied to each patient to obtain a prediction of one year mortality and recalibrated for the local data. This is referred to as the baseline model. A logistic regression model containing the predictions from the Baseline model, the log-base-2 transformed BNP concentrations (as a continuous variable), and the time from BNP measurement to

interRAI-HC assessment was constructed and one-year mortality predictions calculated. This was called the new model.

The added value of BNP to RiskOP was assessed by the relative change in Brier score (Brier skill), difference in area under the receiver operator characteristic curve (δAUC), Integrated Discrimination Improvement (IDI) for those who died ($IDI_{event}$) and those who did not die separately ($IDI_{non-event}$), risk assessment plots (RAP), calibration plots and decision curves. The Brier score is a measure of the variation in risk prediction from the actual outcomes and the Brier skill measures the improvement in this measurement with the addition of a biomarker [17]. The AUC is a measure of discrimination and is the probability that if we were to draw at random a person who died and a person who survived that the risk prediction for the person who died would be greater than for the person who survived. An AUC of 0.5 suggests no discrimination. An AUC of 1 means perfect discrimination. The $IDI_{event}$ represents the mean increase in predicted risk for those who had the event (those who died) whereas the $IDI_{non-event}$ represents the mean decrease in predicted risk for those who did not have the event [18]. RAPs plot Sensitivity verse risk prediction and 1-Specificity verse risk prediction [19]. Improved performance with the addition of a biomarker would be observed by increased separation of the curves. Calibration plots divide the data into quantiles and plot the mean predicted risk verses the actual risk ($n_{event}$/n) within each quantile. Decision curves enable the assessment of the additional net benefit of BNP at specific risk thresholds of relevance to the clinician and patient taking into account both benefits and harms of a proposed treatment [20]. In this case the proposed treatment is simply telling the person that they are at risk of dying in one year, and the benefit of being correct is weighted the same as the harm of being incorrect. The prediction threshold on the x-axis is the threshold of predicted risk above which each person is classified as high risk (in this case high risk of dying in one year) and below which they are classified as low risk. The net benefit is true positives. It is maximal and equal to the prevalence at a prediction threshold of zero. A net benefit of say, 0.1, in this case is equivalent to a strategy that correctly identifies as dying in one year 10 out of 100 at risk people. The higher a decision curve is, the better it performs at identifying true positives.

## Results

There were 18,720 interRAI-HC assessments from 14,713 individuals in the data set. Of these individuals 6,899 (46.9%) had a BNP test within the period of which 1,690 (24.5%) were within 90 days preceding the assessment and had at least one year follow-up, (Fig 1). 1537 individuals had all the required data to calculate one-year mortality predictions with the RiskOP model. The mean age was 82.6 years and 54.2% were female (Table 1). Of these 785 had CHF of whom 324 (41.3%) died within a year. Within those with CHF the median (IQR) BNP was 139 (58–297) pg/mL. Of the 752 without CHF 229 (30.5%) died within one year. Within those without CHF the median (IQR) BNP was 57 (23–130) pg/mL.

Overall there was a modest improvement in prediction with the addition of BNP, (Table 2).

The AUC increased marginally (δAUC = 0.015) from 0.716 to 0.731, and the Brier skill of 1.9% suggests a reduction in overall prediction error. The mean prediction for those who died ($IDI_{event}$) increased marginally (0.9%) and the mean prediction for those who did not die ($IDI_{nonevent}$) decreased marginally (0.6%). In the cohort with CHF, BNP had minimal impact on mortality prediction (Figs 2 and 3. There was no apparent improvement in risk for those who did or who did not die in one year, (Fig 3B). The decision curve suggests a net benefit of the addition of BNP in the mid-prediction threshold range of 0.4 to 0.7, (Fig 4B). In contrast, in those without CHF BNP improved discrimination (δAUC = 0.029 (0.004 to 0.057)) with a final AUC of 0.75. There was no apparent improvement in risk for those who did or did not die in one year, (Fig 3C). The improvement was greatest for those who died with probabilities

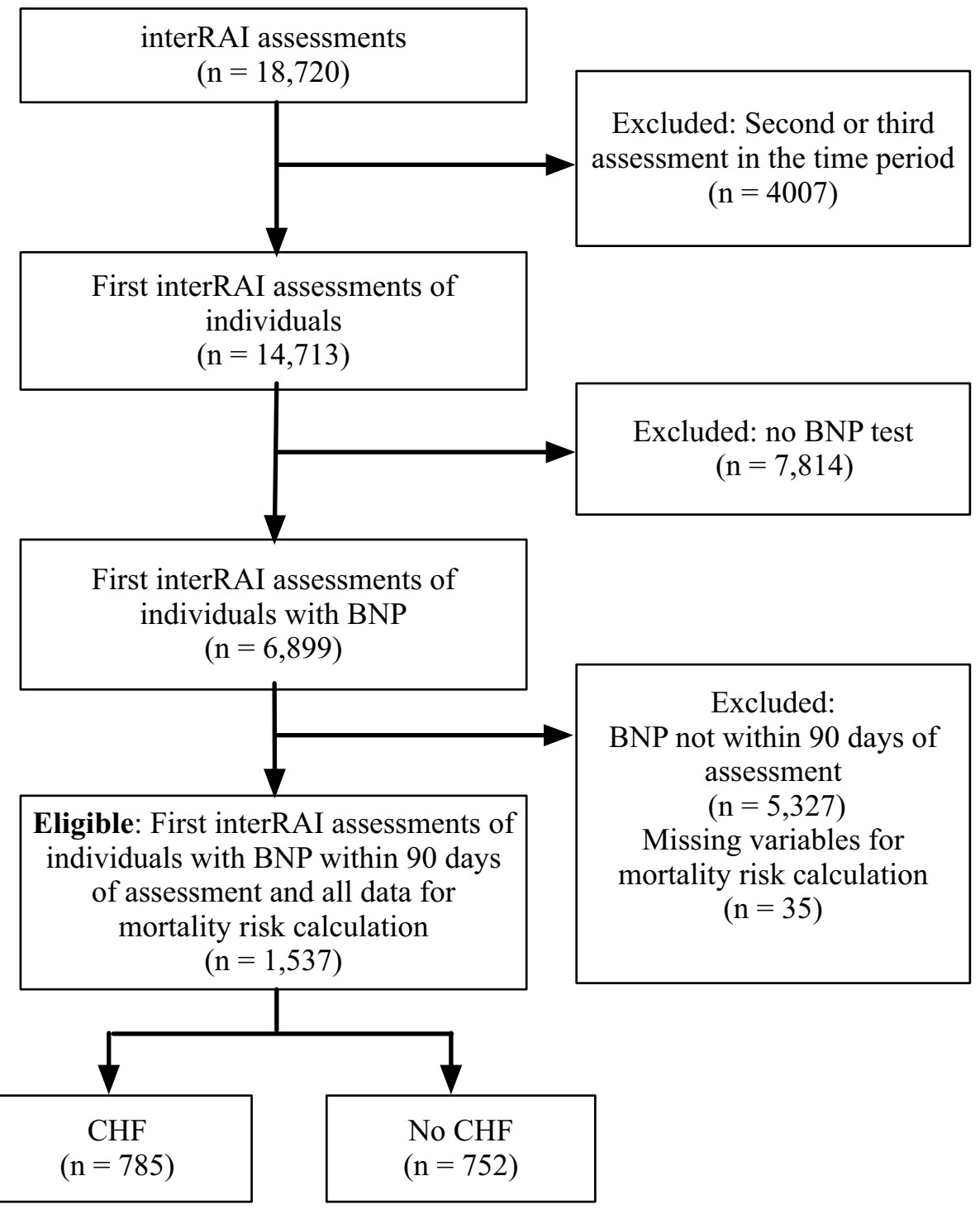

**Fig 1. Consort diagram.**

above about 0.3. The decision curve shows that the addition of BNP improves the application of the risk model amongst those with baseline risk up to 0.75, (Fig 4C). The new model remained well calibrated overall in each of the CHF and no CHF cohorts, (Fig 5).

**Table 1. Demographics.**

| Variable | | Survived (N = 984) | Died 1 (N = 553) | Total (N = 1537) |
|---|---|---|---|---|
| Age | years | 82.0 (8.3) | 83.5 (8.2) | 82.6 (8.3) |
| Sex | Female | 578 (58.4%) | 258 (46.7%) | 833 (54.2%) |
| Ethnicity | Māori | 16 (1.6%) | 21 (3.8%) | 37 (2.4%) |
| | Pacific Peoples | 10 (1.0%) | 8 (1.4%) | 18 (1.2%) |
| | Asian | 4 (0.4%) | 2 (0.4%) | 6 (0.4%) |
| | European | 773 (78.6%) | 428 (77.4%) | 1201 (78.1%) |
| | Other/Unknown/refused to answer/Not Stated | 181 (18.4%) | 94 (17.0%) | 275 (17.9%) |
| Scale BMI | Normal | 304 (30.9%) | 209 (37.8%) | 513 (33.4%) |
| | Underweight | 66 (6.7%) | 63 (11.4%) | 129 (8.4%) |
| | Overweight | 212 (21.5%) | 108 (19.5%) | 320 (20.8%) |
| | Obese | 200 (20.3%) | 66 (11.9%) | 266 (17.3%) |
| | Unknown | 202 (20.5%) | 107 (19.3%) | 309 (20.1%) |
| Cancer | Diagnosed | 129 (13.1%) | 105 (19.0%) | 234 (15.2%) |
| Congestive Heart Failure | Diagnosed | 461 (46.8%) | 324 (58.6%) | 785 (51.1%) |
| Previous stroke | Diagnosed | 196 (19.9%) | 101 (18.3%) | 297 (19.3%) |
| Parkinsons | Diagnosed | 36 (3.7%) | 12 (2.2%) | 48 (3.1%) |
| Dyspnaea | No | 292 (29.7%) | 107 (19.3%) | 399 (26.0%) |
| | Yes-moderate | 277 (28.2%) | 131 (23.7%) | 408 (26.5%) |
| | Yes-normally | 279 (28.4%) | 176 (31.8%) | 455 (29.6%) |
| | Yes-at rest | 136 (13.8%) | 139 (25.1%) | 275 (17.9%) |
| Weight loss | | 149 (15.1%) | 138 (25.0%) | 287 (18.7%) |
| Fatigue | None | 122 (12.4%) | 34 (6.1%) | 156 (10.1%) |
| | Minimal | 365 (37.1%) | 128 (23.1%) | 493 (32.1%) |
| | Moderate | 348 (35.4%) | 217 (39.2%) | 565 (36.8%) |
| | Severe+ | 149 (15.1%) | 174 (31.5%) | 323 (21.0%) |
| Days went out in last 3 days | None | 276 (28.0%) | 255 (46.1%) | 531 (34.5%) |
| | Usually | 73 (7.4%) | 45 (8.1%) | 118 (7.7%) |
| | From 1 to 2d | 334 (33.9%) | 157 (28.4%) | 491 (31.9%) |
| | Three | 301 (30.6%) | 96 (17.4%) | 397 (25.8%) |
| Hours exercise daily | None | 189 (19.2%) | 131 (23.7%) | 320 (20.8%) |
| | Less than 1h | 394 (40.0%) | 241 (43.6%) | 635 (41.3%) |
| | From 1 to 2h | 315 (32.0%) | 144 (26.0%) | 459 (29.9%) |
| | More than 2h | 86 (8.7%) | 37 (6.7%) | 123 (8.0%) |
| On Oxygen therapy | Yes | 35 (3.6%) | 59 (10.7%) | 94 (6.1%) |
| Days since last hospital stay | None in last 90d | 153 (14.2%) | 52 (8.5%) | 205 (12.1%) |
| | From 31 to 90d | 318 (29.5%) | 128 (20.9%) | 446 (26.4%) |
| | From 8 to 30d | 285 (26.4%) | 158 (25.8%) | 443 (26.2%) |
| | From 0 to 7d | 322 (29.9%) | 274 (44.8%) | 596 (35.3%) |
| Change in ADL status in last 6 months | Improved or no change | 455 (46.2%) | 150 (27.1%) | 605 (39.4%) |
| | Declined | 484 (49.2%) | 371 (67.1%) | 855 (55.6%) |
| | Uncertain | 45 (4.6%) | 32 (5.8%) | 77 (5.0%) |

## Discussion

The addition of BNP measurements to a one-year mortality risk prediction model designed for use with a standardised needs assessment for older people marginally added value overall. The additional value was in people without a diagnosis of CHF. More specifically, the study

**Table 2. Discrimination metrics for prediction of mortality.**

| Metric | ALL | CHF | No CHF |
|---|---|---|---|
| n | 1537 | 785 | 752 |
| n died | 554 (519 to 588) | 324 (299 to 352) | 229 (205 to 255) |
| $IDI_{event}$ | 0.009 (0.003 to 0.015) | -0.006 (-0.012 to -0.001) | 0.041 (0.028 to 0.054) |
| $IDI_{non-event}$ | 0.006 (0.002 to 0.01) | 0.003 (-0.002 to 0.009) | 0.014 (0.007 to 0.021) |
| Brier: Baseline | 0.199 (0.19 to 0.208) | 0.216 (0.204 to 0.229) | 0.18 (0.166 to 0.195) |
| Brier: New | 0.195 (0.185 to 0.204) | 0.213 (0.201 to 0.224) | 0.172 (0.158 to 0.188) |
| Brier skill (%) | 1.9 (0.26 to 3.4) | 1.6 (-0.03 to 3.4) | 4.0 (0.68 to 7.6) |
| AUC: Baseline | 0.716 (0.69 to 0.741) | 0.689 (0.651 to 0.726) | 0.725 (0.682 to 0.766) |
| AUC: New | 0.731 (0.705 to 0.757) | 0.699 (0.661 to 0.736) | 0.754 (0.713 to 0.792) |
| δAUC | 0.015 (0.004 to 0.028) | 0.01 (-0.004 to 0.024) | 0.029 (0.004 to 0.057) |

All confidence intervals are 95%.

indicated it had additional prognostic information for clinicians when treating those patients without a diagnosis of CHF and with a risk prediction of below 0.75 (<75% probability of dying within 1-year).

Several previous studies have developed mortality risk prediction models utilising existing health data. These include a large UK study which included a patient questionnaire and bio-chemical tests [21]. In 2020 Canadian research which utilises an earlier version of the inter-RAI-HC, developed the RESPECT tool [22]. Similar to RiskOP, this model was developed in a large group of older adults (mean age 79.6) and had similar discriminatory performance. Given that the interRAI suite of tools are standardised and well established internationally, RESPECT or RiskOP could be expected to be transferrable to other countries using the interRAI-HC.

The large UK based QRISK3 study [6] focussed on future risk of cardiovascular disease. This study utilised the QResearch database at over 1300 General Practices throughout the UK. It included a blood test for high density lipoprotein cholesterol. Adjusted hazard ratios for car-diovascular disease in men was 1.19 (1.18 to 1.19; 95% CI) per unit increase of total cholesterol: HDL cholesterol ratio. It is a rare example of the value of standardised and connected data bases that support not just the individual but population health.

The utilisation of specific blood tests, such as BNP have long been identified as predicting mortality in older adults, including those without any known cardiovascular issues [13]. Indeed, the value of BNP as a predictor of mortality continues to be recognized [13,15,23], including most recently in relation to BNP mortality and its association with Covid-19 [24]. Furthermore, frailty, which is in itself a risk factor for mortality [25], was associated with an increased risk of elevated BNP in a recent study of over 1300 community dwelling elders with no previous cardiovascular history [26].

Given the value of BNP in terms of identifying both frailty and risk of mortality, the addi-tion of this relatively low cost and easy to acquire biomarker to the RiskOP mortality model adds a modest effect to the accuracy of a comprehensive risk prediction, although its value is greater for those without a prior diagnosis of heart failure. That there is little additional prog-nostic value of BNP in those with heart failure may seem surprising. This could simply be that as the people with CHF are more likely to die the covariates in the RiskOP model better corre-late with BNP meaning it has less affect on overall mortality prediction. While intriguing, fur-ther investigation is beyond the scope of this manuscript.

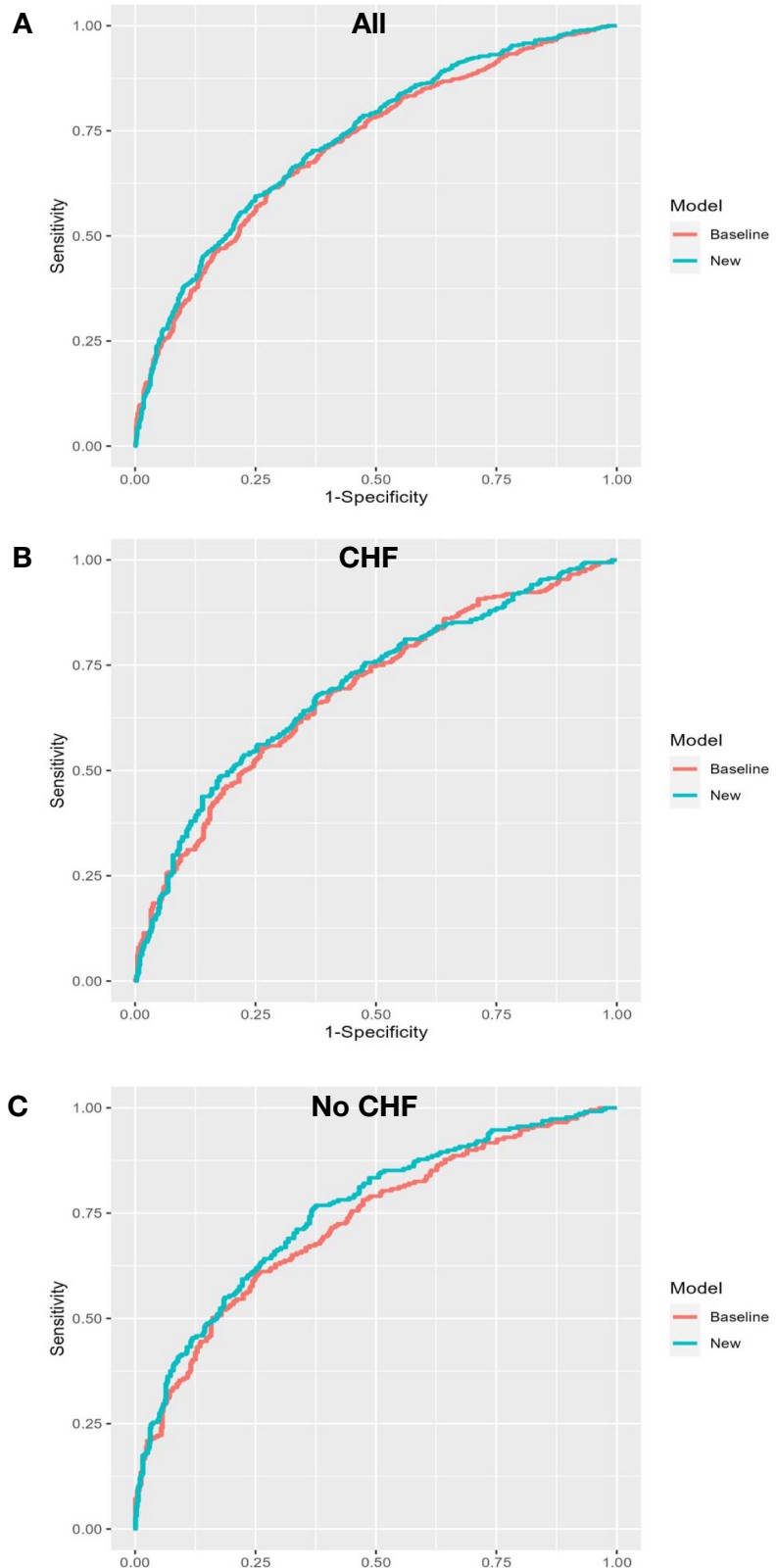

**Fig 2. ROC curves.** Sensitivity verse 1-Specificity. Dotted lines are baseline model RiskOP model, solid lines are the new model with addition of BNP.

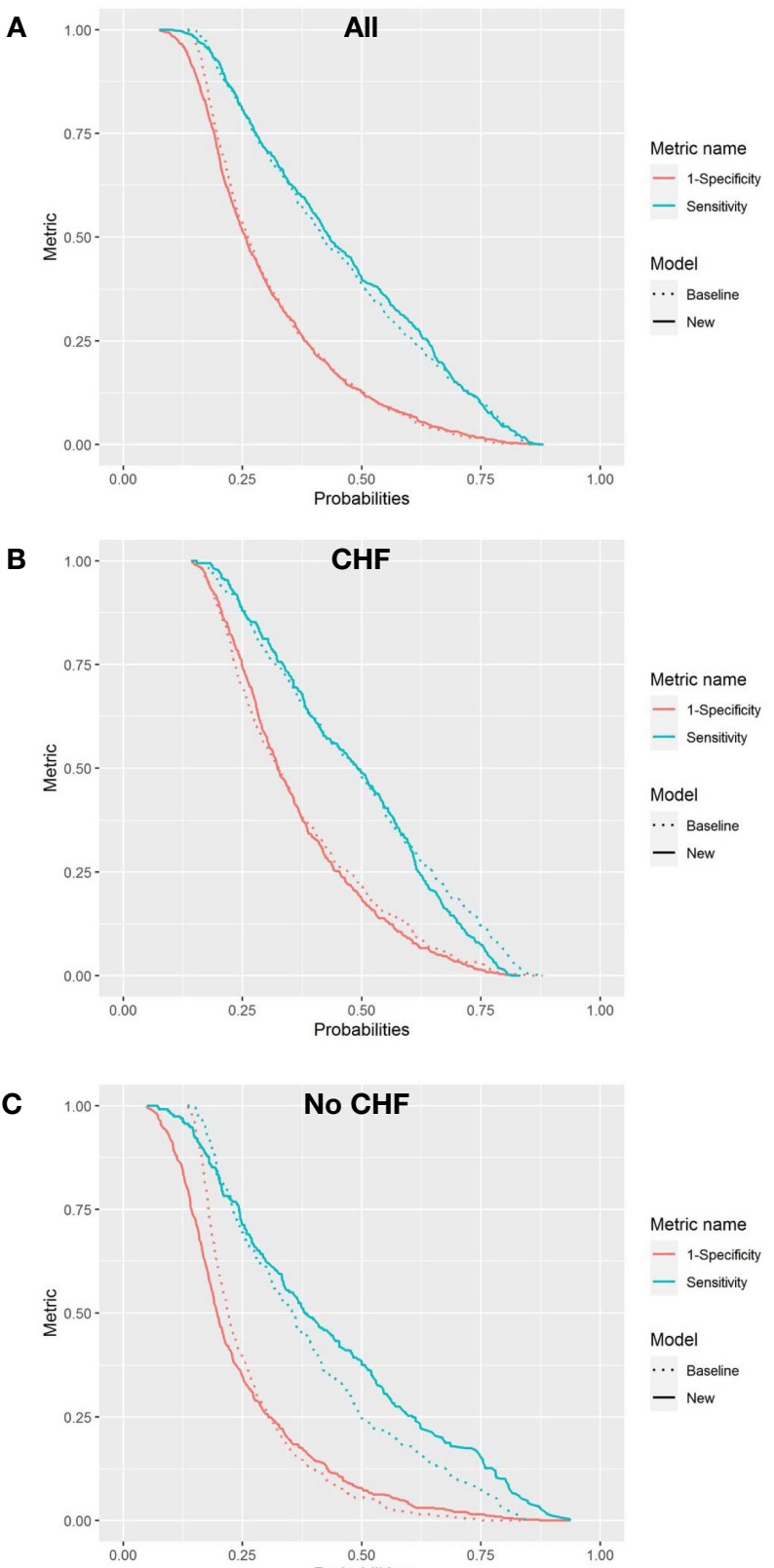

**Fig 3. Risk assessment plots.** Plots illustrating the change in predicted probabilities for participants who died (teal Sensitivity Metric curves) and did not die (red 1-Specificity Metric curves). The dotted line is the predicted probability curve of the baseline model, RiskOP. The solid line is the predicted probability curve with the addition of BNP to RiskOP (New model). Improved prediction is where the teal Sensitivity curve moves towards the upper right corner (higher average risk) with the addition of BNP and/or where the red 1-Specificity curve moves towards the bottom left corner (lower average risk) with the addition of BNP. All patients included in panel A, with separate models for Congestive Heart Failure participants in panel B, and non-Congestive Heart Failure participants in panel C.

Like all assessments, and clinical outcome measures, the prediction models here are designed to be used as a valuable aid to clinical decision making but not as a replacement to skilled clinical judgement and of course individual patient choice which is an important part of any decision-making discussion. Added to this is the important cultural context when any health professional is preparing to discuss a specific patients clinical trajectory and the options available [27]. This group had low numbers of Māori (indigenous New Zealanders) and specific ethnic analysis was not viable. Further research with appropriate numbers of Māori is vital in creating a culturally appropriate model. Additionally, incorporating Haoura Māori values would be evaluated and included before considering the roll-out of any prediction model.

A limitation of this study is that the model has been developed in a national New Zealand based cohort of older adults and the results need to be validated in an international context. However, the advantage of using the interRAI-HC to develop the model is that interRAI-HC is used internationally with similarly trained assessors require ongoing review. This increases the likelihood that this model would validate externally. We also acknowledge that for practical reasons, the interRAI-HC assessments and the BNP blood tests were not usually completed at the same time in the patient journey. Ideally blood draws for biomarker testing would be cotemporaneous with the interRAI assessment. Given many of the interRAI assessors are trained nurses, this is practical. How this cotemporaneous measurement affects the accuracy of the new model would need further investigation. Additionally, while the RiskOP model incorporates Age, Sex, and Body Mass Index, other factors which may be associated with BNP such as inflammation, kidney function, and atrial fibrillation were not measured or included as covariates. If this was able to be done these may well improve the accuracy of the risk prediction. Further investigation would also need to assess performance in sex, age, and ethnicity sub-groups. Finally, the net benefit and decision curve analysis is merely a tool for understanding if BNP could be of added value in decision making. For it to be seen as suitable for clinical practice, a more comprehensive net-benefit analysis would be needed weighing both benefits and harms at each decision point, e.g. for risk stratifying patients to low, intermediate, or high risk.

The addition of a BNP blood test to the RiskOP mortality model improves its accuracy, particularly for those that do not have a diagnosis of congestive heart failure. This is a proof of concept that the addition of a biomarker to risk prediction from a standardised assessment can improve mortality prediction. Further research would need to be undertaken in order to ascertain the value of including additional commonly used blood tests for the older adult population such as sodium, creatinine, albumin, and C-reactive protein. This research would not only need to ascertain the statistical improvement in risk prediction, but also look at a cost-benefit analysis, and ideally would include calibration of models for specific groups including in New Zealand Māori and Pacific Peoples. Improved mortality prediction gives clinicians, patients and family more confidence that they have received the necessary information to make informed decisions about appropriate interventions and treatment options for an individual.

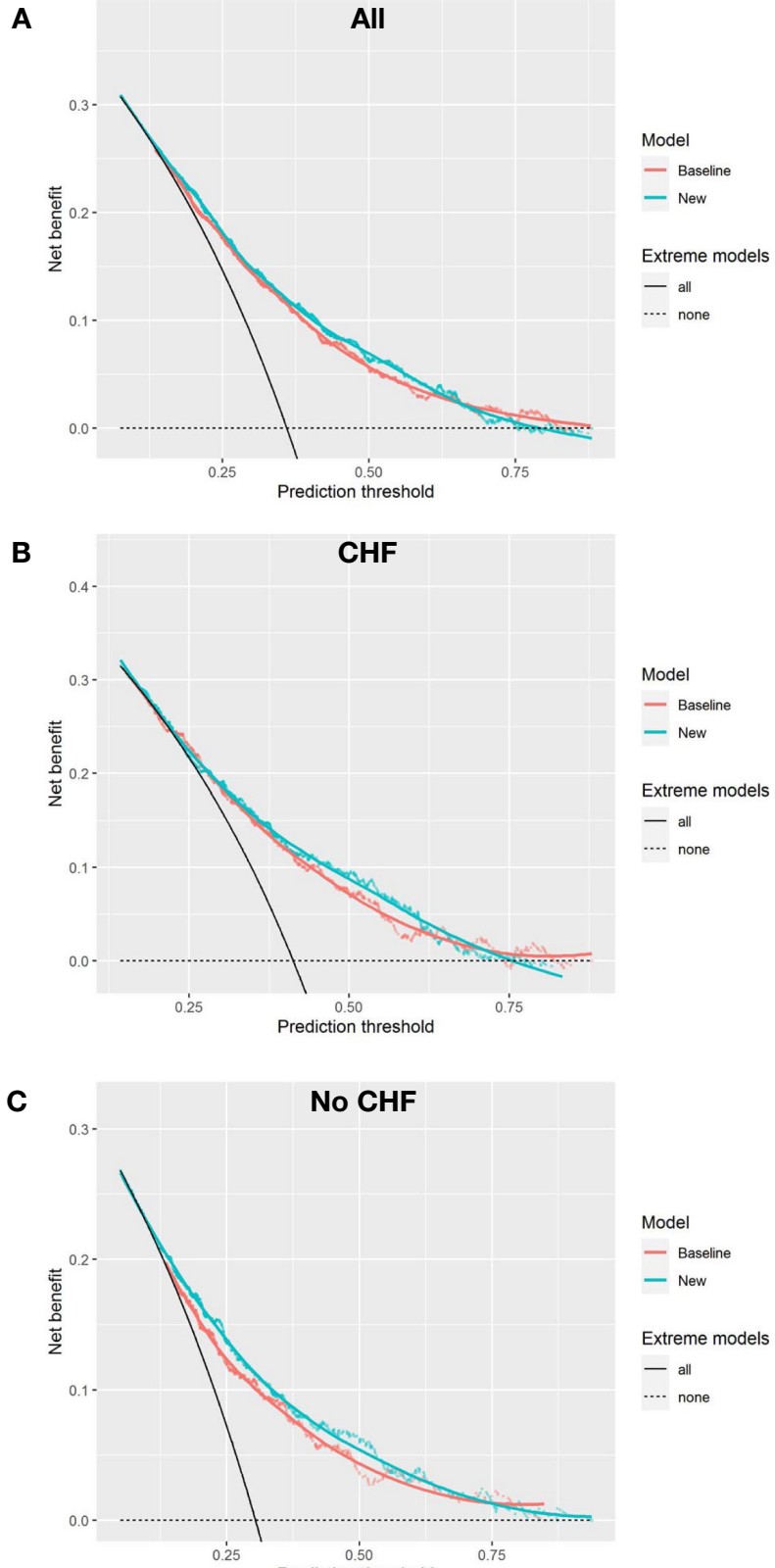

**Fig 4. Decision curves.** At specific prediction of mortality thresholds the new model (teal curve) with the addition of BNP has higher net benefit to the baseline model (red curve). All patients included in panel A, with separate models for Congestive Heart Failure participants in panel B, and non-Congestive Heart Failure participants in panel C.

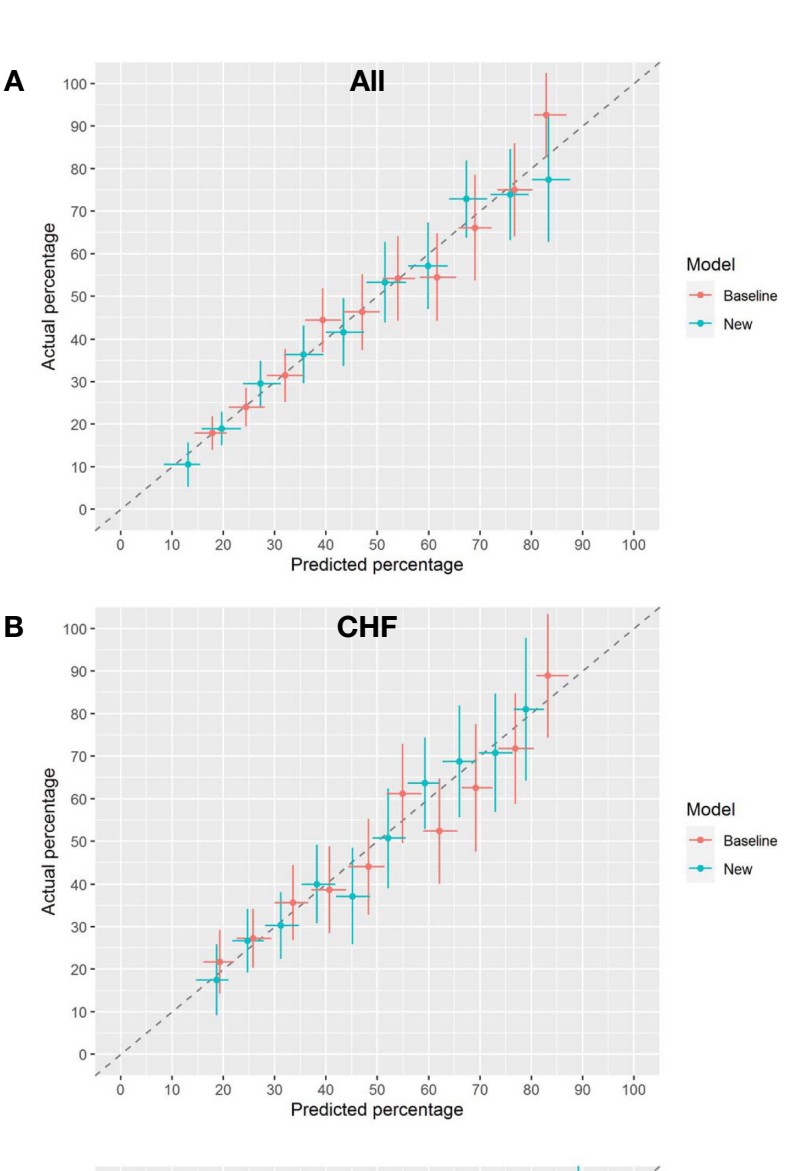

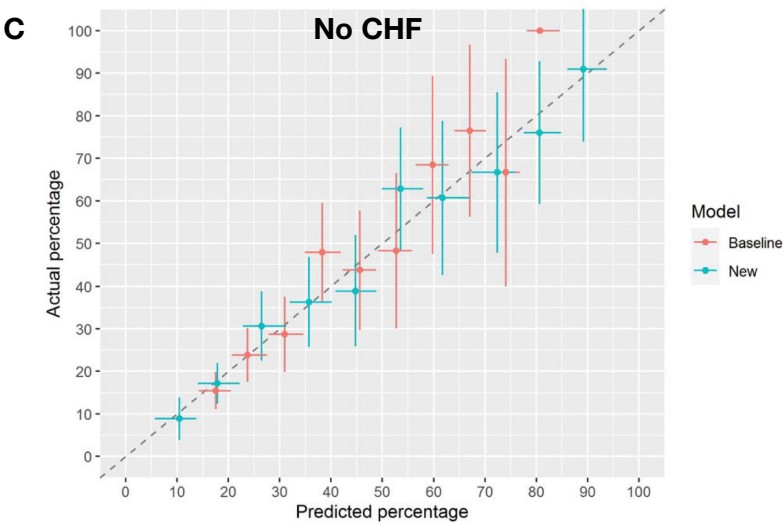

**Fig 5. Calibration curves.** Actual percentage of people with outcomes (and 95% CI) verse predicted percentage (and 95%CI) for all patients split into 10 groups. Perfect calibration is the dashed line. The new model (teal curve) has similar or better calibration than the baseline model (red curve). All patients are included in panel A. Panel B is for participants with Congestive Heart Failure, and Panel C for participants without Congestive Heart Failure.

## Author Contributions

**Conceptualization:** John W. Pickering, Hamish A. Jamieson.

**Data curation:** John W. Pickering.

**Formal analysis:** John W. Pickering.

**Funding acquisition:** Hamish A. Jamieson.

**Investigation:** Richard Scrase, Richard Troughton, Hamish A. Jamieson.

**Methodology:** John W. Pickering, Richard Troughton.

**Writing – original draft:** John W. Pickering, Hamish A. Jamieson.

**Writing – review & editing:** John W. Pickering, Richard Scrase, Richard Troughton, Hamish A. Jamieson.

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
