## [Decision Letter · Decision Letter 0]

9 Jun 2022

PONE-D-22-10241Evaluation of the added value of Brain Natriuretic Peptide to a validated mortality risk-prediction model in older people using a standardised international clinical assessment toolPLOS ONE

Dear Dr. Pickering,

Thank you for submitting your manuscript to PLOS ONE. After careful consideration, we feel that it has merit but does not fully meet PLOS ONE’s publication criteria as it currently stands. Therefore, we invite you to submit a revised version of the manuscript that addresses the points raised during the review process.

We look forward to receiving your revised manuscript.

Kind regards,

Juan A López-Rodríguez

Academic Editor

PLOS ONE

Journal Requirements:

Reviewers' comments:

Reviewer's Responses to Questions

**Comments to the Author**

1. Is the manuscript technically sound, and do the data support the conclusions?

Reviewer #1: Yes

2. Has the statistical analysis been performed appropriately and rigorously? 

Reviewer #1: I Don't Know

3. Have the authors made all data underlying the findings in their manuscript fully available?

Reviewer #1: No

4. Is the manuscript presented in an intelligible fashion and written in standard English?

Reviewer #1: Yes

5. Review Comments to the Author

Reviewer #1: Evaluation of the added value of Brain Natriuretic Peptide to a validated mortality risk prediction model in older people using a standardised international clinical assessment tool

The authors explore the question if addition of the cardiac biomarker BNP adds prognostic value to a established mortality prediction score in elderly adults in NZ.

This is a well-designed study with a well-phenotyped cohort and fair sample size. Improving risk prediction in the elderly (mean age >80 in this cohort) is of crucial clinical importance for the patient, patient’s family and physician. The main findings of this study are (1) adding BNP to the RiskOP model significantly improved the accuracy of one-year mortality prediction, (2) especially in patients without a diagnosis of CHF. The study utilizes advanced statistical models.

Major

BNP or NT-proBNP is a valuable prognostic biomarker in a wide range of cardio-vascular diseases. BNP is influenced by age, BMI, inflammation, gender, atrial fibrillation and kidney function. The authors should adjust for those metrics before performing survival analyses.

The authors should explain how BNP levels were incorporated in the RiskOP Model. Were age appropriate thresholds used?

It would be helpful for the reader to report the actual BNP values of the cohorts.

The authors should explain why they chose BNP as a biomarker, why not serum sodium, albumin or CRP?

The mean age was >80 years old, individuals at this age have a very high change for diastolic dysfunction. Do the authors have access to echocardiograms in the CHF cohort? Is there a way to report the prevalence of Afib in both cohorts?

The authors should clarify when how the RiskOP model was derived, is this part of the interRAI assessment?

Are the authors suggesting that BNP should be part of a routine assessment in the elderly? Could they estimate a benefit-cost ratio?

Image quality of the figures needs to be improved

Minor

page 9, 69, sentence needs improved syntax

page 11, 102, doe the authors mean “informed” instead of “inform”?

page 12, 146, what do the authors mean by “cost effective”?

page 17, 236, this is confusing. So how many BNP measurements were included in the analysis?

Page 17, table 1 the column Died should have the figure 1 deleted

Page 19, 258 should read “died” not “die”

Page 19, 268, Fig2, should read “died” not “dies”

Page 20, 283 and 284, should read congestive heart failure

Page 20, 290, sentence needs improved syntax

6. PLOS authors have the option to publish the peer review history of their article (what does this mean?). If published, this will include your full peer review and any attached files.

Reviewer #1: **Yes: **Nils Patrick Nickel

---

## [Author Response · Author response to Decision Letter 0]

8 Jul 2022

PONE-D-22-10241 Response to Reviewers

Evaluation of the added value of Brain Natriuretic Peptide to a validated mortality risk-prediction model in older people using a standardised international clinical assessment tool

Thank you Dr Nickel for your considered review. We have responded under each point below.

Reviewers' comments:

Reviewer's Responses to Questions

Comments to the Author

1. Is the manuscript technically sound, and do the data support the conclusions?

Reviewer #1: Yes

Response: Thank you.

2. Has the statistical analysis been performed appropriately and rigorously? 

Reviewer #1: I Don't Know

Response: As the first author, and an experienced biostatistican, I appreciate the honesty of this response. 

3. Have the authors made all data underlying the findings in their manuscript fully available?

Reviewer #1: No

Response: Because of restrictions of use of clinical data we cannot make it freely available. However, we can make it available if contacted subject to ethics approval from a New Zealand ethics board and a formal use agreement between ourselves and those who wish to use the data. For New Zealand data this is particularly important because some of the data is from Māori, the indigenous people of New Zealand, who are acknowledged to have sovereignty of their data and its use.

4. Is the manuscript presented in an intelligible fashion and written in standard English?

Reviewer #1: Yes

Response: Thank you.

5. Review Comments to the Author

Reviewer #1: Evaluation of the added value of Brain Natriuretic Peptide to a validated mortality risk prediction model in older people using a standardised international clinical assessment tool

The authors explore the question if addition of the cardiac biomarker BNP adds prognostic value to a established mortality prediction score in elderly adults in NZ.

This is a well-designed study with a well-phenotyped cohort and fair sample size. Improving risk prediction in the elderly (mean age >80 in this cohort) is of crucial clinical importance for the patient, patient’s family and physician. The main findings of this study are (1) adding BNP to the RiskOP model significantly improved the accuracy of one-year mortality prediction, (2) especially in patients without a diagnosis of CHF. The study utilizes advanced statistical models.

Major

BNP or NT-proBNP is a valuable prognostic biomarker in a wide range of cardio-vascular diseases. BNP is influenced by age, BMI, inflammation, gender, atrial fibrillation and kidney function. The authors should adjust for those metrics before performing survival analyses.

The authors should explain how BNP levels were incorporated in the RiskOP Model. Were age appropriate thresholds used?

Response: Age, Sex, and BMI are part of the RiskOP model (see end of the 1st paragraph of the methods section) , therefore from a statistical analysis perspective it is not appropriate to account for them a second time (i.e. their influence, through RiskOp, is already accounted for). Unfortunately, we do not have information on AF and kidney function. We have acknowledged this limitation in the discussion:

 “Additionally, which the RiskOP model incorporates Age, Sex, and Body Mass Index, other factors which may be associated with BNP such as inflammation, kidney function, and atrial fibrillation were not measured or included as covariates. If this was able to be done these may well improve the accuracy of the risk prediction.“

BNP concentrations were included as a continuous variable. Dichotomising at any threshold would effectively throw away information and make the models poorer. We have added the phrase “(as a continuous variable)” in the description of the model at the end of the second paragraph of the statistical analysis section.

It would be helpful for the reader to report the actual BNP values of the cohorts.

The authors should explain why they chose BNP as a biomarker, why not serum sodium, albumin or CRP?

Response: Thank you for the suggestion. We have added the information on BNP concentrations to the first paragraph of the Results. We chose BNP simply because it is one of the most frequently measured biomarkers in Heart Failure patients in New Zealand. We thought this adequate to assess the proof of concept that the addition of a biomarker to risk prediction from a standardised assessment can improve mortality prediction (line 347). Future studies may look at multi-biomarker response and we have now introduced this possibility into the discussion, we had already indicated that sodium or albumin may be included, but have now added CRP and creatinine.

The mean age was >80 years old, individuals at this age have a very high change for diastolic dysfunction. Do the authors have access to echocardiograms in the CHF cohort? Is there a way to report the prevalence of Afib in both cohorts?

Response: Unfortunately, we do not have access to echocardiograms or AF information. We note that in this cohort of people with high-needs, in New Zealand echo-cardiograms are often not done. We have acknowledged the limitation in the limitations section (as in the addition we made in response to an earlier point raised by the referee).

The authors should clarify when how the RiskOP model was derived, is this part of the interRAI assessment?

Response: The development of the RiskOP model was published separately with detail on how in Lancet eClinical Medicine (reference 11). It would be too much detail to include here. We have clarified in the text that it was derived on people with a Home Care interRAI assessment (interRAI-HC) – i.e. the same as all people in this present study. The sentence in the introduction now reads:

 “The authors recently developed and validated a mortality risk model (RiskOP) for those people aged 65 plus using data from the interRAI-HC only[11].”

And we added a sentence in the methods:

 “The RiskOP model had been developed in a subset of these patients (up to March 2018)[11].”

Are the authors suggesting that BNP should be part of a routine assessment in the elderly? Could they estimate a benefit-cost ratio?

Response: No, we are not suggesting it should be part of a routine assessment in all elderly. Any conclusion we draw must be limited to the high-needs cohort (in this case identified by needing an interRAI Home Care assessment). What we are demonstrating in this proof-of-concept is that a biomarker, in this case BNP, could make risk prediction for mortality better. It is too early to suggest routine adoption, the consequences and benefits of better risk prediction need to be discussed by the medical community. Additionally, we need to assess the added value of other biomarkers, such as creatinine and CRP, which could be measured at the same time. We believe only then a cost-benefit analysis would be applicable (we cannot estimate with on the current data). We have expanded the discussion around future directions:

 “Further research would need to be undertaken in order to ascertain the value of including additional commonly used blood tests for the older adult population such as sodium, creatinine, albumin, and C-reactive protein. This research would not only need to ascertain the statistical improvement in risk prediction, but also look at a cost-benefit analysis, and ideally would include calibration of models for specific groups including in New Zealand Māori and Pacific Peoples.”

Image quality of the figures needs to be improved

Response: Thank you – probably a product of the pdf generation, we will work with the editors to ensure good quality.

Minor

Response: Thank you. Unfortunately, these line and page numbers do not correspond with the ones on our documents, nevertheless, we have tracked down these as best we can.

page 9, 69, sentence needs improved syntax

Response: We were uncertain on the exact sentence referred to, but revised the entire page and found a sentence with a syntax issue which we fixed.

page 11, 102, doe the authors mean “informed” instead of “inform”?

Response: the sentence is “...to make informed decisions...” and is therefore correct.

page 12, 146, what do the authors mean by “cost effective”?

Response: Simply that its costs are reasonable compared to other costs (ie it is inexpensive). We have rephrased this to avoid confusion.

page 17, 236, this is confusing. So how many BNP measurements were included in the analysis?

Response: Just one per participant, we have added the word “single” in the methods to clarify and changed the wording of the sentence in question.

Page 17, table 1 the column Died should have the figure 1 deleted

Response: Thank you, deleted.

Page 19, 258 should read “died” not “die”

Response: Thank you, corrected.

Page 19, 268, Fig2, should read “died” not “dies”

Response: Thank you, corrected.

Page 20, 283 and 284, should read congestive heart failure

Response: Oops – thank you (I must now check the work I’m doing at the same time with CAD that I’ve not made the opposite error there!).

Page 20, 290, sentence needs improved syntax

Response: We have reworded.

---

## [Decision Letter · Decision Letter 1]

4 Sep 2022

PONE-D-22-10241R1Evaluation of the added value of Brain Natriuretic Peptide to a validated mortality risk-prediction model in older people using a standardised international clinical assessment toolPLOS ONE

Dear Dr. Pickering,

Thank you for submitting your manuscript to PLOS ONE. After careful consideration, we feel that it has merit but does not fully meet PLOS ONE’s publication criteria as it currently stands. Therefore, we invite you to submit a revised version of the manuscript that addresses the points raised during the review process.

The article is very interesting and certainly adds to scientific knowledge.  It is a well-designed study regarding the addition of a biomarker to existing prediction tool. Methods and results are very well written. However, the discussion does not address appropriately the limitations of the modified tool and the explanation for the lack of utility in those with CHF. Although the improvement is statistically significant, it seems not to be very relevant from a clinical point of view. The conclusions are a bit misleading overeating the value of the modified tool and should be modified stating that the improvement in performance is just marginal.

It should be discussed that the addition of a laboratory examination would be a major change in a score that is intended to be solely based on clinical data. Although the authors mentioned that is an inexpensive marker, this is not the same for most countries. In fact, BNP is unavailable even in reference hospitals in less developed countries.

As mentioned, there is no explanation about the reason why to perform a subgroup analysis on those patients with and without CHF. Moreover, there is a lack of discussion of the possible reason of the poor performance of the modified tool in the CHF subgroup. I think that BNP probably does not add prognostic information if the patient has already a diagnosis of CHF. In patients without CHF, BNP probably identify some undiagnosed or subclinical cases. This or other possible explanations should be included in a separate paragraph in the discussion.

We look forward to receiving your revised manuscript.

Kind regards,

Alonso Soto, PhD

Academic Editor

PLOS ONE

Reviewers' comments:

Reviewer's Responses to Questions

**Comments to the Author**

1. If the authors have adequately addressed your comments raised in a previous round of review and you feel that this manuscript is now acceptable for publication, you may indicate that here to bypass the “Comments to the Author” section, enter your conflict of interest statement in the “Confidential to Editor” section, and submit your "Accept" recommendation.

Reviewer #1: All comments have been addressed

Reviewer #2: (No Response)

Reviewer #3: (No Response)

2. Is the manuscript technically sound, and do the data support the conclusions?

Reviewer #1: Yes

Reviewer #2: Yes

Reviewer #3: Yes

3. Has the statistical analysis been performed appropriately and rigorously? 

Reviewer #1: Yes

Reviewer #2: No

Reviewer #3: Yes

4. Have the authors made all data underlying the findings in their manuscript fully available?

Reviewer #1: Yes

Reviewer #2: Yes

Reviewer #3: Yes

5. Is the manuscript presented in an intelligible fashion and written in standard English?

Reviewer #1: Yes

Reviewer #2: No

Reviewer #3: Yes

6. Review Comments to the Author

Reviewer #1: Excellent paper! Very interesting for the readership. The authors did an excellent job in answering the reviewers questions.

Reviewer #2: 1) If the authors have adequately addressed your comments raised in a previous round of review …

I had the opportunity to read the questions of the first reviewer and the answers of the authors. To my knowledge, they answered correctly to the remarks. The manuscript was adapted correctly.

2) Is the manuscript technically sound, and do the data support the conclusions?

Absolutely.

3) Has the statistical analysis been performed appropriately and rigorously?

Appropriately: this article might be interesting for clinicians. But no single clinician understands more than 10% of the statistics applied. The question is, if so many analyses are necessary for this relatively simple question. "μηδέν άγαν" (meden agan)! Never too much, said Socrates. So why all these analyses, while a simple ROC says enough in this case. Moreover, the promised basic ROC curve (graph) is missing.

Rigorously: decision curve analysis: crucial is here the balance between giving erroneously a bad message, or erroneously a good message. The authors boldly put this balance at 1/1, which is against the philosophy of medical decision making. The key task in decision analysis is always the estimation of this balance. Without a thorough even qualitative research concerning this balance, decision curve analysis makes no sense.

4) Have the authors made all data underlying the findings in their manuscript fully available?

This question has been answered in the first review.

5) Is the manuscript presented in an intelligible fashion and written in standard English

The English is perfect. But I think even some statisticians will have difficulty understanding all analyses done. I had to go back to the difficult decision curve analysis, to understand fully what the authors mean.

As the authors state that this article might be interesting for clinicians, I propose to show the basic ROC curves with AUC statistics. And to state clearly the difference between significance and effect size. The latter is very small in this research, even in non-cardiac patients. The authors use the word ‘marginally’ in the first paragraph of the discussion, but go further with ‘clear added value’.

6) Abstract

Methods: the authors state: “Incremental value was assessed by change in Area Under the Receiver Operating Characteristic Curve (AUC)”

In the results section we (clinicians) miss a classical ROC curve. Instead, we see graphs that take time to understand.

Conclusions: I did not understand immediately the conclusions. The authors conclude that the BNP improves the prediction for non-cardiac patients. I think I’m not the only reader to expect BNP to influence prediction especially in cardiac patients. I would state this as an unexpected paradox: BNP has an added value only in non-CHF.

Suggestion:

After revision by the first reviewer, I should state this is an excellent article, but not for clinicians or Plos One. In the present writing, I would submit it to Medical Decision Making, given the thorough statistical analyses (except for the decision curve analysis, which should be revised). If it is intended for Plos One, I would revise it along the suggestions I made hereabove.

Reviewer #3: In this manuscript, Pickering et al. performed a proof-of-concept study to evaluate the potential of adding BNP to the one-year mortality prediction model. The study included >1500 individuals with completed interrail-HC assessment and BNP measurement. It was found that the addition of BNP would increase AUC by 0.015 (95% CI 0.004-0.028). The improvement mostly came from individuals without CHF. Overall, the manuscript is well-written.

Following are some minor comments.

1. In the abstract, please define CHF. Given that more than half of individuals had CHF, such information could be also included in the abstract.

2. Please also include the number of mortalities in the abstract

3. The sample size used for the calculation is unclear. In Table 2, it is 1537, but in the Results section, it is indicated that 1572 individuals had all the required data. The total number of individuals with CHF (n=785) and without CHF (n=752) would be 1537. This number should be also used in the abstract, but not 1690 individuals.

4. It would be useful to comment on why the addition of BNP only improved the prediction performance for individuals without CHF but not for those with CHF.

5. Is there any sex-specific difference in the prediction performance?

7. PLOS authors have the option to publish the peer review history of their article (what does this mean?). If published, this will include your full peer review and any attached files.

Reviewer #1: **Yes: **Nils Patrick Nickel

Reviewer #2: **Yes: **Jef Van den Ende

Reviewer #3: No

---

## [Editor Report · Decision Letter 2]

4 Nov 2022

Evaluation of the added value of Brain Natriuretic Peptide to a validated mortality risk-prediction model in older people using a standardised international clinical assessment tool

PONE-D-22-10241R2

Dear Dr. Pickering,

We’re pleased to inform you that your manuscript has been judged scientifically suitable for publication and will be formally accepted for publication once it meets all outstanding technical requirements.

Kind regards,

Alonso Soto, PhD

Academic Editor

PLOS ONE

---

## [Editor Report · Acceptance letter]

10 Nov 2022

PONE-D-22-10241R2 

Evaluation of the added value of Brain Natriuretic Peptide to a validated mortality risk-prediction model in older people using a standardised international clinical assessment tool. 

Dear Dr. Pickering:

I'm pleased to inform you that your manuscript has been deemed suitable for publication in PLOS ONE. Congratulations! Your manuscript is now with our production department. 

Kind regards, 

on behalf of

Dr. Alonso Soto 

Academic Editor

PLOS ONE